# Transcript assembly and annotations: Bias and adjustment

**Qimin Zhang**[1], **Mingfu Shao**[1,2]*

**1** Department of Computer Science and Engineering, School of Electrical Engineering and Computer Science, The Pennsylvania State University, University Park, Pennsylvania, United States of America, **2** Huck Institutes of the Life Sciences, The Pennsylvania State University, University Park, Pennsylvania, United States of America

* mxs2589@psu.edu

## Abstract

Transcript annotations play a critical role in gene expression analysis as they serve as a reference for quantifying isoform-level expression. The two main sources of annotations are RefSeq and Ensembl/GENCODE, but discrepancies between their methodologies and information resources can lead to significant differences. It has been demonstrated that the choice of annotation can have a significant impact on gene expression analysis. Furthermore, transcript assembly is closely linked to annotations, as assembling large-scale available RNA-seq data is an effective data-driven way to construct annotations, and annotations are often served as benchmarks to evaluate the accuracy of assembly methods. However, the influence of different annotations on transcript assembly is not yet fully understood. We investigate the impact of annotations on transcript assembly. Surprisingly, we observe that opposite conclusions can arise when evaluating assemblers with different annotations. To understand this striking phenomenon, we compare the structural similarity of annotations at various levels and find that the primary structural difference across annotations occurs at the intron-chain level. Next, we examine the biotypes of annotated and assembled transcripts and uncover a significant bias towards annotating and assembling transcripts with intron retentions, which explains above the contradictory conclusions. We develop a standalone tool, available at https://github.com/Shao-Group/irtool, that can be combined with an assembler to generate an assembly without intron retentions. We evaluate the performance of such a pipeline and offer guidance to select appropriate assembling tools for different application scenarios.

**Data Availability Statement:** The EN10 dataset used in this study are downloaded from the ENCODE project, consisting of 10 paired-end RNA-seq samples. Their SRA accession IDs are SRR307903, SRR307911, SRR315323,

## Author summary

Transcript annotations are essential foundations for transcriptomic studies, offering valuable insights into gene structures, functions, and acting as references for isoform-level expression expression quantification and differential analysis. However, the impact of different annotations on transcript assembly remains uncertain. We demonstrated that the choice of an annotation can lead to conflicting outcomes when evaluating assemblers. Our investigation revealed the distinctive features of annotations that led to the

SRR315334, SRR387661, SRR534291, SRR534307, SRR534319, SRR545695, and SRR545723. The HS7 dataset consists of 7 paired-end RNA-seq samples downloaded from the Long Read Genome Annotation Assessment Project (https://www.gencodegenes.org/pages/LRGASP/). Their accession IDs are ENCFF766OAK paired with ENCFF644AQW, ENCFF198RQU paired with ENCFF620HBM, ENCFF247XJT paired with ENCFF785SWH, ENCFF201EVI paired with ENCFF591ISP, ENCFF221SLD paired with ENCFF223VFL, ENCFF145IIO paired with ENCFF597GZT, and ENCFF701OIK paired with ENCFF139HIY. All relevant data are within the manuscript and its Supporting information files.

**Funding:** This work is supported by the US National Science Foundation (DBI-2019797 to MS and DBI-2145171 to MS) and the US National Institutes of Health (R01HG011065 to MS). The funders had no role in study design, data collection and analysis, decision to publish, or preparation of the manuscript.

**Competing interests:** The authors have declared that no competing interests exist.

aforementioned contradictory conclusion, through a comprehensive comparison of annotations from the perspectives of biotypes and gene structures, contributing to a broader, deeper understanding of annotations. Our research provides guidance in making well-informed choice of annotations and assemblers for practical RNA-seq data analysis.

# 1 Introduction

The isoform-level expression analysis has become a common toolbox in biological and bio-medical studies. This analysis generally involves quantifying the expression levels of annotated transcripts given the RNA-seq data, followed by statistical methods to identify differentially expressed transcripts. Popular tools used in this pipeline include RSEM [1], kallisto [2], Salmon [3], edgeR [4], and DESeq2 [5], among others. The transcriptome, which is the collection of annotated transcripts, plays a crucial role in the analysis, as it serves as a reference for isoform quantification and splicing quantification [6–8]. The construction of high-quality, reliable, and complete transcriptomes for model species has been a long-standing community effort. Currently, RefSeq [9], led by NCBI, and Ensembl/GENCODE [10], led by EMBL-EBI, are the two main sources of annotations, with updates and curations constantly being made.

It is widely acknowledged that the RefSeq and Ensembl annotations differ significantly due to differences in methodology and information resources. Generally, RefSeq annotation prioritizes experimental evidence, while Ensembl annotation incorporates more computational predictions and includes more novel splicing variants [11]. The choice of an annotation depends on the specific need; for example, RefSeq is commonly used for variant studies [12], while Ensembl annotations are preferred for extensive research initiatives such as ENCODE [13], gnomAD [14], and GTEx [15]. Several studies have investigated the impact of different annotations on gene expression analysis. It has been reported that the choice of annotation has a significant effect on RNA-seq read mapping, gene/isoform quantification, and differential analysis [16–18]. In addition, integrating diverse annotations can markedly improve transcriptomic and genetic studies [19].

Computational methods are increasingly used to identify novel isoforms to complement annotations of model species and to construct transcriptomes for non-model species, thanks to the availability of large-scale deposited RNA-seq data. The process of reconstructing full-length transcripts from RNA-seq reads, known as transcript assembly, has been extensively studied, with significant progress made in advancing the theory [20–22] and in developing practical assemblers, including Cufflinks [23], CLASS2 [24], TransComb [25], FLAIR [26], StringTie [27], Scallop [28], StringTie2 [29], and Scallop2 [30], to just name a few. There is a close relationship between transcript assembly and annotations. On one hand, transcript assembly offers a data-driven venue to construct annotations [31]; on the other hand, annotations serve as a ground-truth to evaluate assemblers on real RNA-seq data where the true expressed transcripts are unknown.

We study the impact of annotations on transcript assembly in this work. We evaluate the accuracy of two recent and popular assemblers Scallop2 and StringTie2 with different annotations (Section 2.1). Surprisingly, we found that Scallop2 performs better than StringTie2 when evaluated with Ensembl annotations but worse with RefSeq annotations. To uncover the underlying reasons, we first systematically compare the structural similarity between different annotations to investigate the primary sources of divergence (Section 2.2). We found that while annotations already differ significantly at the intron-exon boundary and junction levels, the differences are most pronounced at the intron-chain level. We then investigate if the

differences are related to transcript biotypes (Section 2.3). We observed that transcripts with intron retentions contribute the most significant disparity to RefSeq and Ensembl annotations. Meanwhile, Scallop2 and StringTie2 also behave differently in assembling such transcripts. We therefore conclude that the joint bias in assembling and annotating transcripts with intron retentions leads to the opposite evaluation results. Finally, we propose criteria and develop a tool to adjust the biases in intron retention for an assembly and provide guidance for a suitable pipeline based on testing the assemblers with and without using the adjustment (Section 2.4).

## 2 Results

### 2.1 Evaluating assemblers with different annotations

We show that divergent conclusions can be drawn when transcript assemblers are evaluated using different annotations. We investigate this phenomenon by comparing two widely-used reference-based assemblers, StringTie2 and Scallop2, with five transcriptome annotations derived from two human genome build, GRCh38 and T2T-CHM13 [32], on 17 paired-end RNA-seq samples from two datasets, 10 samples from EN10 and 7 samples from HS7, aligned with two popular aligners, STAR [33] and HISAT2 [34]. The assembly accuracies are evaluated using GffCompare [35]. Details about the comparison of the methods, accuracy measures, and evaluation pipeline are provided in Section 4.1. We do not intend to conduct a comprehensive benchmarking analysis for assemblers but rather to reveal the divergence of annotations and their impacts on evaluating assemblers. Please see Section 2.5 for a broader comparison under more experimental settings and our insights about the assemblers' methodological differences.

Table 1 compares the accuracy of StringTie2 and Scallop2. As shown in the table, Scallop2 outperforms StringTie2 when evaluated with Ensembl and CHM13 annotations, evidenced by Scallop2 outperforming on all samples (68 and 34, respectively; using adjusted precisions to break ties). However, a different conclusion is reached when evaluated with RefSeq annotations, as StringTie2 outperforms on more samples (38 out of 68) than Scallop2. The high level of agreement between Ensembl and CHM13 annotations (from T2T-CHM13 genome build) is expected since they are very similar, as illustrated in Section 2.2. We therefore focus on exploring the differences between Ensembl and RefSeq annotations.

We further demonstrate the dramatic discrepancy of RefSeq and Ensembl annotations in evaluating transcript assemblers, by comparing their accuracies on individual samples. The results are shown in Fig 1 (using GRCh38 annotations) and Fig 2 (using T2T-CHM13 annotations). Across all combinations, Scallop2's accuracy is significantly higher under Ensembl annotation than under RefSeq. Conversely, the pattern is almost reversed for StringTie2, with its accuracy evaluated under Ensembl being either lower than that under RefSeq in the case of GRCh38 annotations Fig 1), or only slightly higher in the case of T2T-CHM13 annotations Fig 2).

### 2.2 Comparison of structural similarities

The results presented in Section 2.1 clearly highlight the substantial differences between RefSeq and Ensembl annotations. This prompts us to investigate the primary sources of these divergences. In particular, since a transcript can be represented as a chain of alternating exons and introns, we seek to determine the level of transcript structure that contributes the most to these differences, whether it be at the individual exon-intron boundary, the junction (pair of intron boundaries), or the chain of junctions. To address this question, we propose several metrics and use them to evaluate the similarity of annotations at these three levels. Details about the metrics definitions are provided in Section 4.2.

**Table 1. Comparison of the assembly accuracy, measured with precision (%) and the number of matching transcripts, of StringTie2 and Scallop2 using different annotations as the reference.** In each combination (of dataset, aligner, genome build, annotation) the two metrics are averaged over all samples in the dataset. Symbol ⟨⟩ indicates that one method gets higher on one metric but lower on the other; symbol > indicates that StringTie2 outperforms on both metrics, while < indicates Scallop2 outperforms on both metrics. The three columns of *raw counts* give the number of samples in each category by comparing raw precision and recall. Samples in the ⟨⟩ category are further compared using the adjusted precision, and the number of samples are merged into either > or < category accordingly, shown in the two columns of under *adjusted*.

| dataset | aligner | genome | annotation | StringTie2 | | Scallop2 | | raw counts | | | adjusted | |
|---|---|---|---|---|---|---|---|---|---|---|---|---|
| | | | | prec. | # mat. | prec. | # mat. | > | ⟨⟩ | < | > | < |
| EN10 | HISAT2 | GRCh38 | RefSeq | 32.1% | 14906 | 28.8% | 16256 | 0 | 8 | 2 | 1 | 9 |
| EN10 | HISAT2 | T2T | RefSeq | 28.2% | 13298 | 25.3% | 14396 | 0 | 8 | 2 | 1 | 9 |
| EN10 | STAR | GRCh38 | RefSeq | 34.3% | 15113 | 30.1% | 14929 | 6 | 3 | 1 | 8 | 2 |
| EN10 | STAR | T2T | RefSeq | 30.2% | 13279 | 27.4% | 14416 | 0 | 8 | 2 | 1 | 9 |
| HS7 | HISAT2 | GRCh38 | RefSeq | 41.5% | 18523 | 37.5% | 18395 | 7 | 0 | 0 | 7 | 0 |
| HS7 | HISAT2 | T2T | RefSeq | 36.6% | 16429 | 33.1% | 16358 | 6 | 1 | 0 | 7 | 0 |
| HS7 | STAR | GRCh38 | RefSeq | 42.2% | 18695 | 41.9% | 18300 | 5 | 2 | 0 | 7 | 0 |
| HS7 | STAR | T2T | RefSeq | 37.1% | 16425 | 37.2% | 16250 | 3 | 4 | 0 | 6 | 1 |
| | Summary | | RefSeq | 35.3% | 15834 | 32.7% | 16163 | 27 | 34 | 7 | 38 | 30 |
| EN10 | HISAT2 | GRCh38 | Ensembl | 32.2% | 14684 | 32.7% | 18205 | 0 | 4 | 6 | 0 | 10 |
| EN10 | HISAT2 | T2T | Ensembl | 29.1% | 13707 | 29.9% | 17100 | 0 | 3 | 7 | 0 | 10 |
| EN10 | STAR | GRCh38 | Ensembl | 32.7% | 14412 | 34.2% | 18133 | 0 | 4 | 6 | 0 | 10 |
| EN10 | STAR | T2T | Ensembl | 31.5% | 13885 | 32.9% | 17406 | 0 | 4 | 6 | 0 | 10 |
| HS7 | HISAT2 | GRCh38 | Ensembl | 38.8% | 17304 | 38.6% | 18971 | 0 | 5 | 2 | 0 | 7 |
| HS7 | HISAT2 | T2T | Ensembl | 37.1% | 16619 | 36.8% | 18165 | 0 | 5 | 2 | 0 | 7 |
| HS7 | STAR | GRCh38 | Ensembl | 39.7% | 17594 | 43.7% | 19145 | 0 | 0 | 7 | 0 | 7 |
| HS7 | STAR | T2T | Ensembl | 38.3% | 16939 | 42.0% | 18355 | 0 | 0 | 7 | 0 | 7 |
| | Summary | | Ensembl | 34.9% | 15643 | 36.4% | 18185 | 0 | 25 | 43 | 0 | 68 |
| EN10 | HISAT2 | T2T | CHM13 | 29.5% | 13926 | 29.9% | 17101 | 0 | 4 | 6 | 0 | 10 |
| EN10 | STAR | T2T | CHM13 | 31.7% | 13940 | 32.7% | 17332 | 0 | 4 | 6 | 0 | 10 |
| HS7 | HISAT2 | T2T | CHM13 | 37.5% | 16832 | 37.0% | 18236 | 0 | 5 | 2 | 0 | 7 |
| HS7 | STAR | T2T | CHM13 | 38.4% | 16965 | 41.9% | 18273 | 0 | 0 | 7 | 0 | 7 |
| | Summary | | CHM13 | 34.3% | 15416 | 35.4% | 17736 | 0 | 13 | 21 | 0 | 34 |

We plotted the Jaccard similarities across different annotations in Fig 3. It clearly shows that the Ensembl and CHM13 annotations of the T2T-CHM13 genome build are highly similar, with Jaccard of 0.91 at the boundary and junction levels, and 0.80 at the intron-chain level. However, the Ensembl and RefSeq annotations in both genome builds exhibit significant divergence, with Jaccard values lower than 0.69 and 0.57 at the boundary and junction levels, respectively. This difference is most pronounced at the intron-chain level, where the Jaccard similarity drops to 0.19, indicating that the intron-chain is the primary contributor to the structural disparity between Ensembl and RefSeq annotations.

Next, we measure the similarity of individual genes in different annotations, aiming to determine whether the difference of annotations can be attributed to a small portion of genes. The method to construct the correspondence between genes in two annotations and definitions of Jaccard similarities of every constructed gene pair at boundary, junction, and intron-chain levels, are provided in Section 4.3. The gene pairs constructed with this approach aligns very well with gene nomenclature. Specifically, there are 24957 gene pairs between RefSeq and Ensembl annotations according to the HUGO Gene Nomenclature. Among these, 23478 pairs (94.1%) can be found in the gene pairs constructed with our approach.

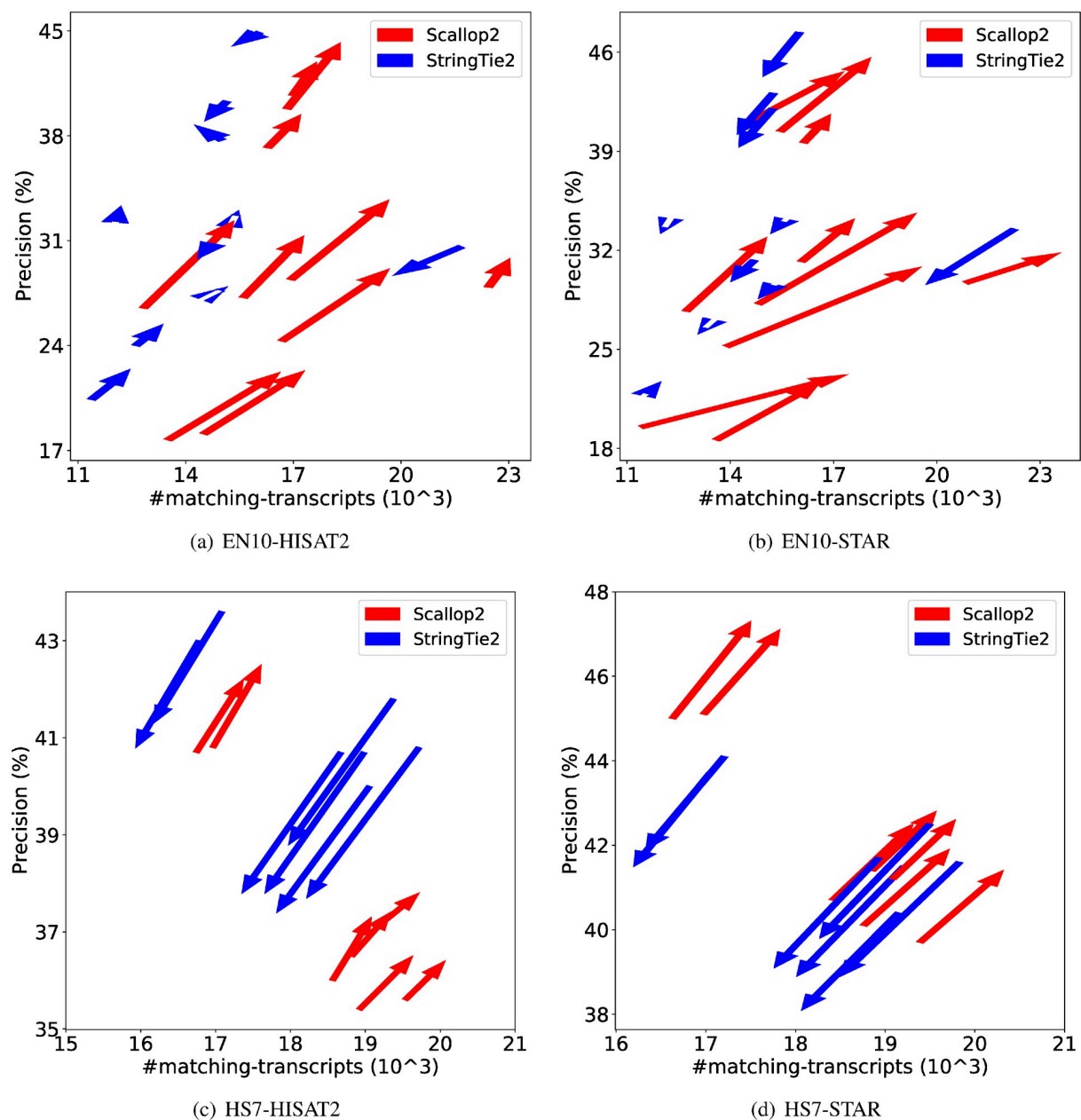

**Fig 1. Illustrating the difference of assembly accuracy evaluated with RefSeq and Ensembl annotations, both from GRCh38 genome build.** Each arrow represents a sample, pointing from the accuracy evaluated with RefSeq annotation to that with Ensembl annotation. The subfigures correspond to the four combinations of dataset (EN10 or HS7) and aligner (HISAT2 or STAR).

The distribution of the Jaccard similarities are shown in Fig 4. As before, we observe that Ensembl and CHM13 annotations in the T2T-CHM13 genome build are almost identical to each other. However, Ensembl and RefSeq annotations in either genome build show significant divergence, especially at the intron-chain level. Furthermore, the majority of the gene pairs between Ensembl and RefSeq annotations are divergent, as evidenced by the quartiles in Fig 4.

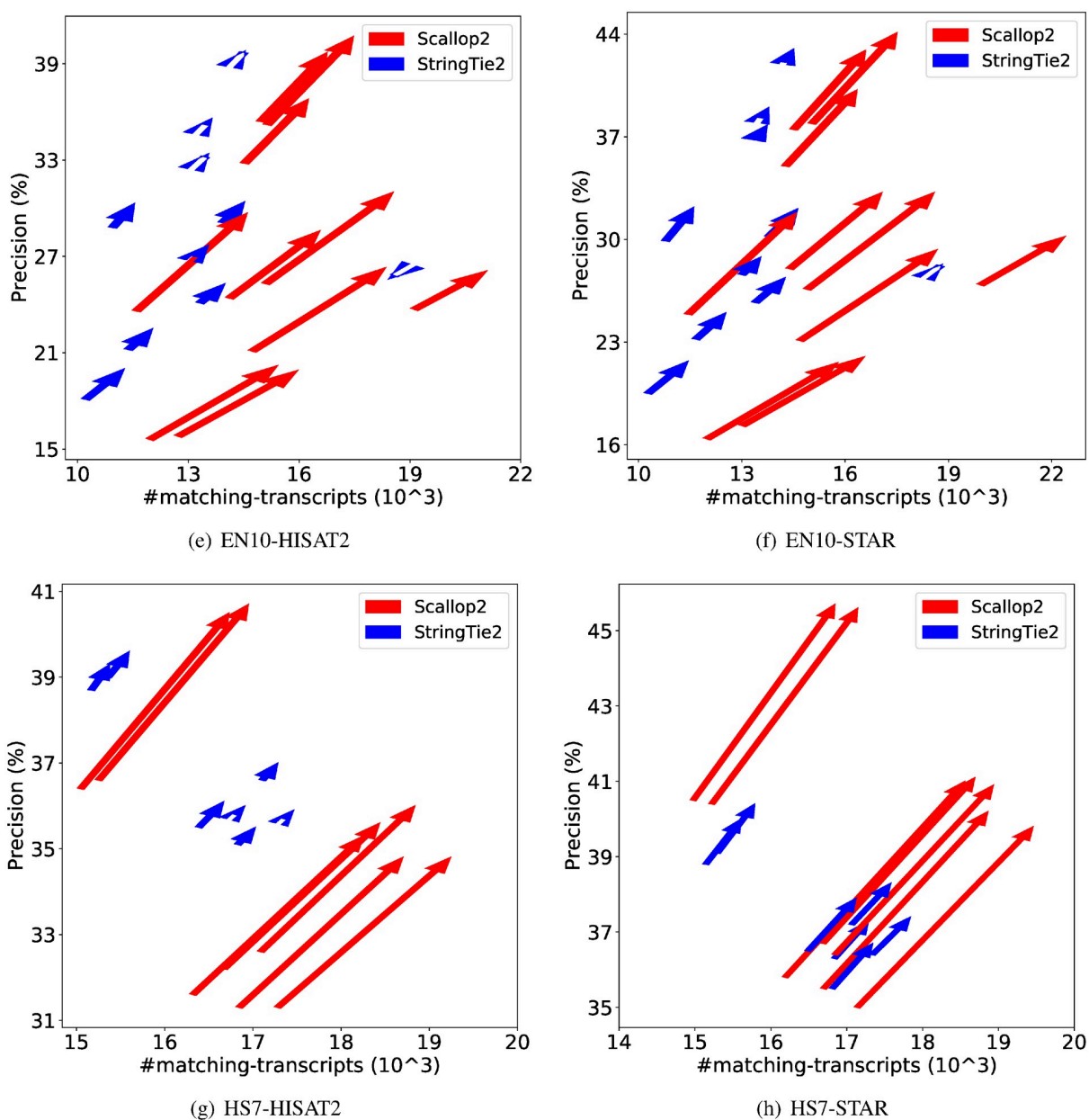

**Fig 2. Illustrating the difference of assembly accuracy evaluated with RefSeq and Ensembl annotations, both from T2T-CHM13 genome build.** Each arrow represents a sample, pointing from the accuracy evaluated with RefSeq annotation to that with Ensembl annotation. The subfigures correspond to the four combinations of dataset (EN10 or HS7) and aligner (HISAT2 or STAR).

## 2.3 Comparison of transcript biotypes

The Ensembl annotation provides each annotated transcript with a "biotype" that indicates its biological category and function. By leveraging this information, we aim to investigate whether different annotations or assemblers show bias towards specific biotypes.

In Table 2, we compare the distribution of multi-exon transcripts belonging to different biotypes between the Ensembl and RefSeq annotations. As biotype information is not available in the RefSeq annotation, we report the number and percentage of transcripts annotated by

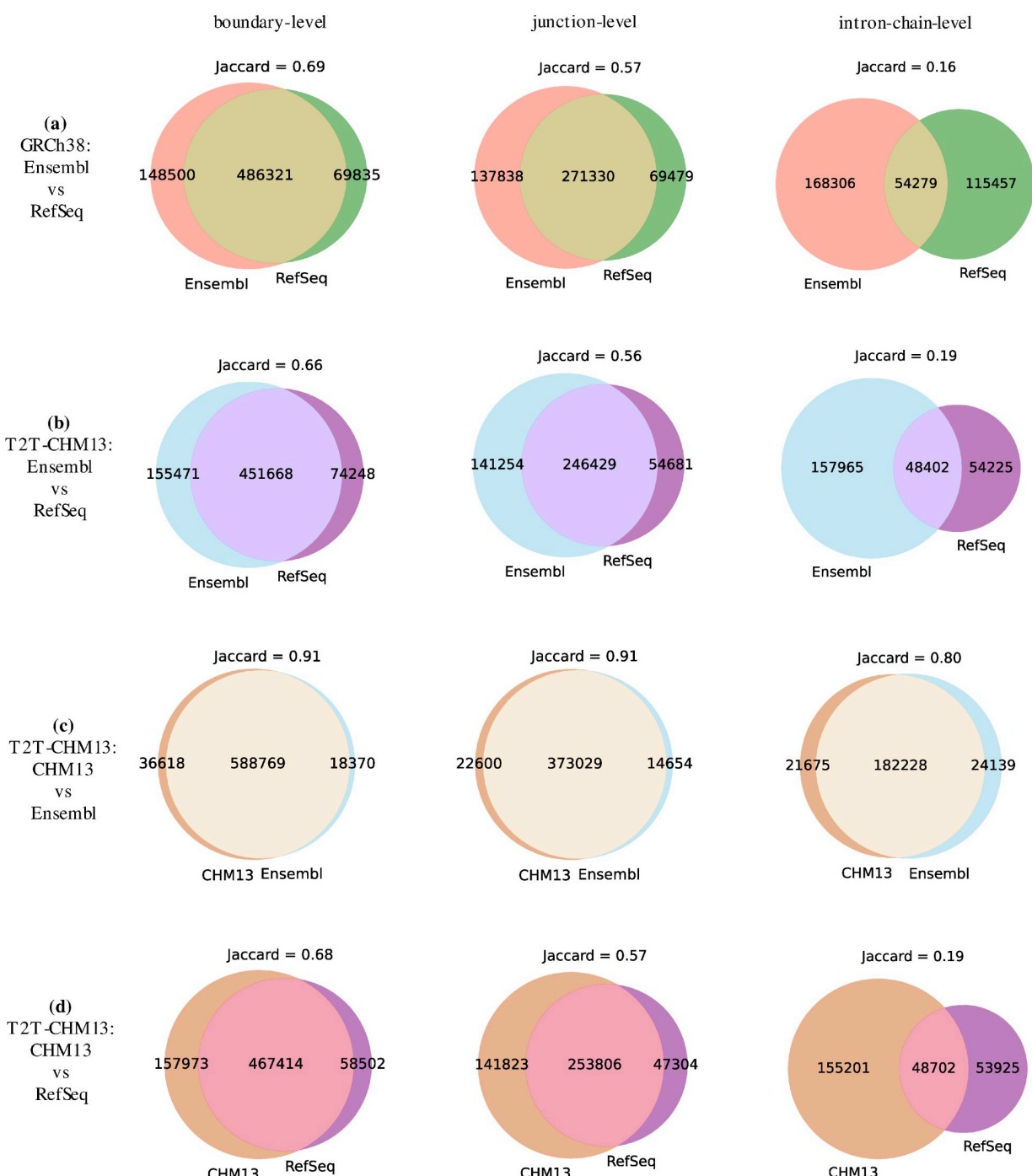

**Fig 3. The Jaccard similarity of 4 pairs of annotations (4 subfigures) at the level of boundary, junction, and intron-chain.**

Ensembl in each biotype that are also annotated in RefSeq. (Two transcripts are considered the same if they share the same intron-chain.) Our analysis reveals huge divergences in several biotypes, such as "retained_intron", "processed_transcript", and "processed_pseudogene", where only a tiny portion of them are annotated in RefSeq. Of particular interest is the "retained

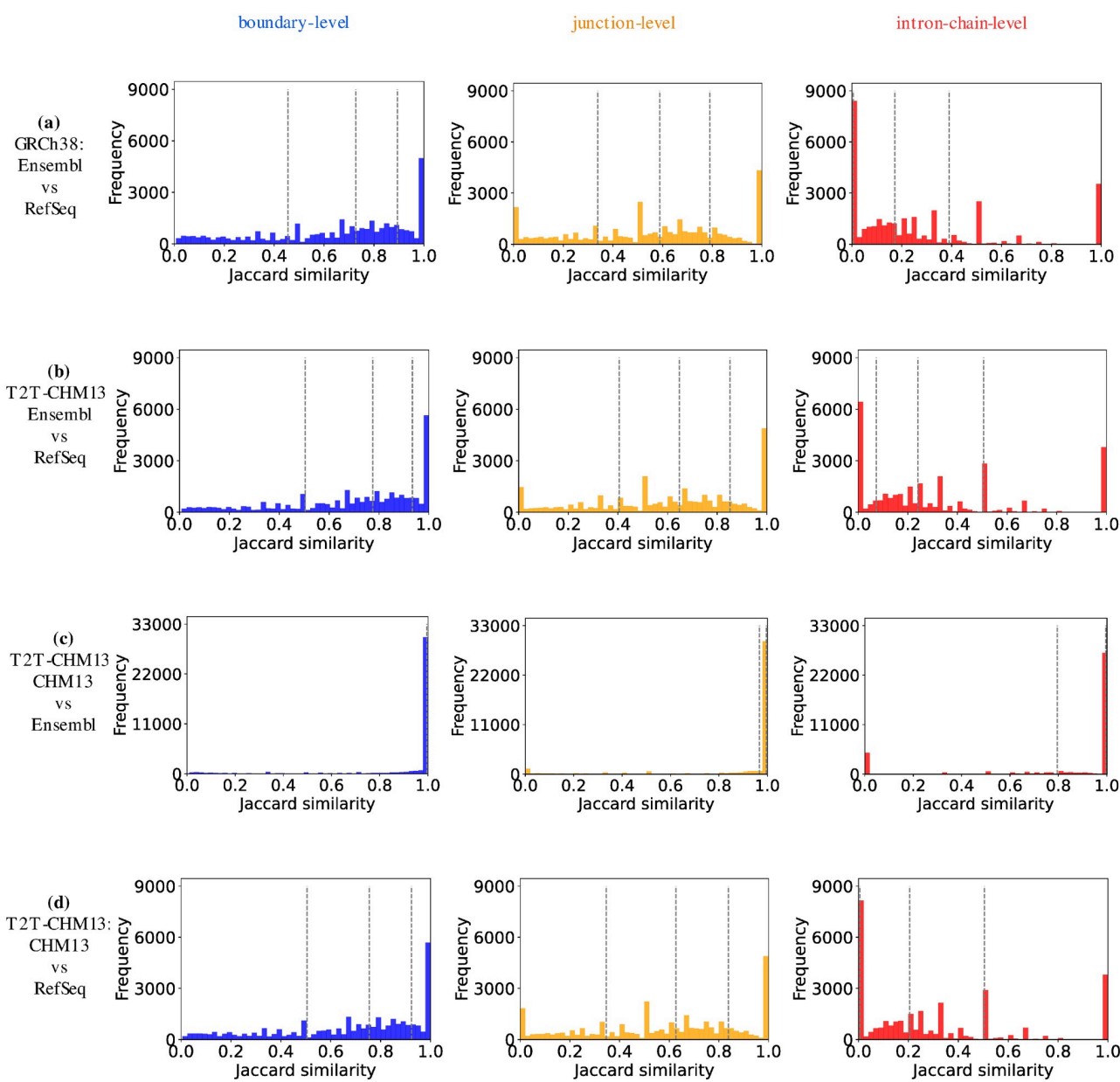

**Fig 4. The distribution of Jaccard similarities of all paired genes in each pair of compared annotations at the level of boundary, junction, and intron-chain.** The three dashed lines in each subfigure mark the Jaccard similarity at the 25th, 50th, and 75th percentile of the total frequency.

intron" biotype, which is the third largest biotype in the Ensembl annotation, but only 1.3% of them appear in the RefSeq annotation.

We investigate whether the dissimilarities between Scallop2 and StringTie2 when evaluated with Ensembl and RefSeq annotations (as discussed in Section 2.1) can be attributed to differences in biotypes. We classify the matching transcripts based on their biotypes, using the Ensembl annotation as the reference. The comparison of the five largest biotypes is presented in Fig 5. Notably, Scallop2 identifies significantly more matching transcripts in the "retained intron" biotype compared to StringTie2. We therefore conjecture that it is the bias towards

**Table 2. Illustration of the number of multi-exon transcripts in different biotypes between Ensembl and RefSeq annotations (GRCh38 build).** The first column lists biotypes defined by the Ensembl annotation; the second column lists the number of multi-exon transcripts in each biotype; the third and the fourth columns give the number and the percentage of multi-exon transcripts in each biotype that are also annotated in the RefSeq annotation.

| Ensembl transcript biotype | # annotated in Ensembl | # annotated in RefSeq | percentage (%) |
|---|---|---|---|
| protein_coding | 86333 | 41848 | 48.5 |
| lncRNA | 47848 | 7058 | 14.8 |
| retained_intron | 32782 | 411 | 1.3 |
| processed_transcript | 30592 | 1884 | 6.2 |
| nonsense_mediated_decay | 20754 | 2991 | 14.4 |
| unprocessed_pseudogene | 1421 | 21 | 1.5 |
| processed_pseudogene | 1294 | 1 | 0.1 |
| transcribed_unprocessed_pseudogene | 737 | 18 | 2.4 |
| IG_V_gene | 142 | 124 | 87.3 |
| transcribed_unitary_pseudogene | 120 | 5 | 4.2 |
| TR_V_gene | 105 | 101 | 96.2 |
| IG_V_pseudogene | 100 | 35 | 35.0 |
| non_stop_decay | 97 | 8 | 8.2 |
| transcribed_processed_pseudogene | 58 | 1 | 1.7 |
| protein_coding_LoF | 51 | 31 | 60.8 |
| unitary_pseudogene | 43 | 1 | 2.3 |
| TR_V_pseudogene | 23 | 20 | 87.0 |
| artifact | 19 | 11 | 57.9 |
| pseudogene | 19 | 0 | 0.0 |
| IG_C_gene | 18 | 9 | 50.0 |
| TEC | 7 | 0 | 0.0 |
| TR_C_gene | 6 | 4 | 66.7 |
| IG_C_pseudogene | 2 | 1 | 50.0 |
| translated_unprocessed_pseudogene | 1 | 0 | 0.0 |

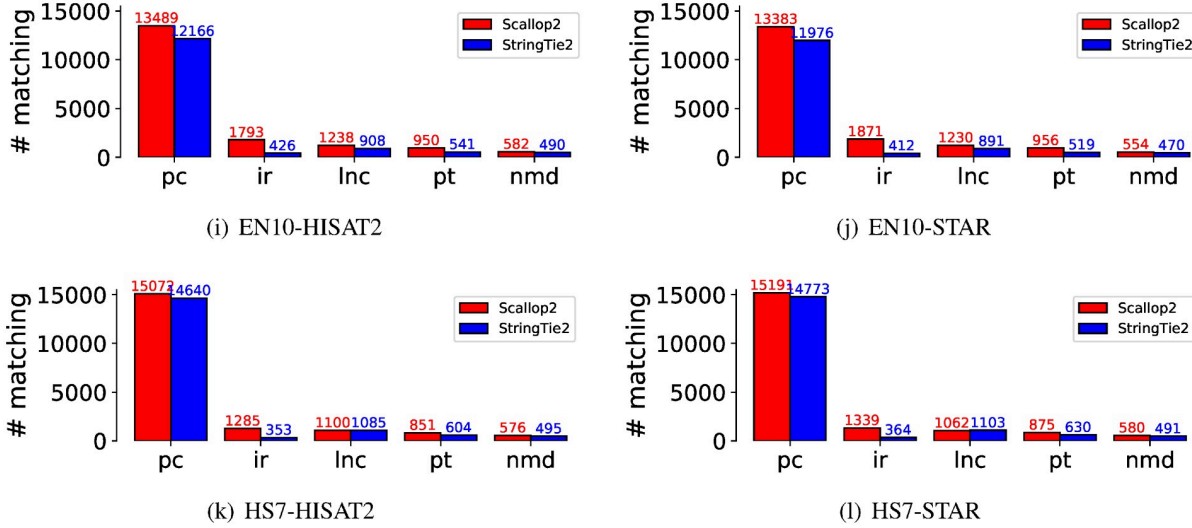

**Fig 5. Comparison of the matching transcripts assembled by StringTie2 and Scallop2 in the five largest biotypes using the Ensembl annotation as the reference.** The average number over all samples in each dataset is reported in the barplot. The 5 biotypes are: protein_coding (pc), retained_intron (ir), lncRNA (lnc), processed_transcript (pt), and nonsense_mediated_decay (nmd). These five biotypes account for 98.6% of the total transcripts in Ensembl.

retained-intron transcripts, i.e., Scallop2 assembles more such transcripts than StringTie2 while Ensembl annotates more such transcripts than RefSeq, that is the primary reason behind the opposite conclusion when StringTie2 and Scallop2 are evaluated against the two annotations.

To further support the above hypothesis, we analyzed the number of assembled transcripts that were annotated in Ensembl but not in RefSeq. These transcripts are considered true positives by Ensembl and false positives by RefSeq. The results for the five largest biotypes are presented in Fig 6, which shows that the largest difference is due to the "retained intron" biotype. Therefore, we conclude that the discrepancy in evaluation between Scallop2 and StringTie2 is caused by the differences in transcripts with intron retentions between Ensembl and RefSeq annotations.

To seek evidence for supporting the intron-retained transcripts annotated in Ensembl, we investigate a long-read dataset used in the LRGASP [36]. The dataset was sequenced with the CapTrap PacBio protocol on a mixed sample prepared from the human H1 ES/Definitive Endoderm cell line. This dataset was used in the Challenge 1 of LRGASP, aiming to compare different computational methods (and sequencing platform and library prep approaches, etc) on assembling transcripts with a high-quality genome. We directly use the assemblies (e.g., assembled full-length transcripts) by two participating methods, namely Bambu and String-Tie2. We investigate the overlap between each assembly and the Ensembl transcripts with intron retentions. Specifically, we define an annotated intron-retained transcript in Ensembl to be *validated* if its intron-chain coordinates exactly matches one transcript in an assembly. This can be done by calling GffCompare. By using the Bambu assembly, we found that 3566 (10.9% of the total annotated 32782 intron-retained transcripts in Ensembl) can be validated, and this number is 4773 (14.6% of the annotated ones) when validating using the StringTie2 assembly. We note that the Bambu and StringTie2 assemblies contain 30369 and 46866 transcripts, respectively; the percentages of matching intron-retained transcripts in Ensembl are 11.7% and 10.2% for Bambu and StringTie2 respectively, which are at the same level of the percentage of intron-retained transcripts in Ensembl (32782 of 222572, 14.7%). In contrast, as

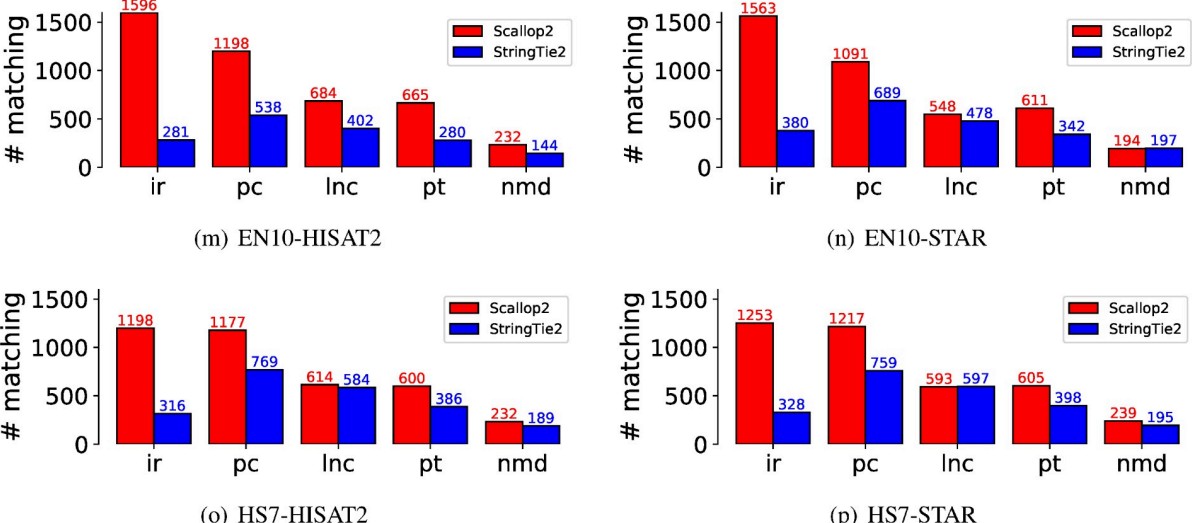

**Fig 6. Comparison of the transcripts assembled by Scallop2 and StringTie2 that are annotated by Ensembl but not by RefSeq in each of the five largest biotypes.** The average number over all samples in each dataset is reported in the barplot. The 5 biotypes are ir: retained_intron, pc: protein_coding, lnc: lncRNA (long non-coding RNA), pt: processed_transcript, nmd: nonsense_mediated_decay.

shown in Table 2, RefSeq annotates 411 transcripts with intron-retentions. Considering that thousands of Ensembl transcripts with intron-retentions that match assemblies from long-reads, it is plausible that RefSeq might be too conservative in annotating transcripts with intron-retentions.

## 2.4 Quantifying the impact of intron retentions

We hence investigate intron retentions. We first establish a formal definition for (partial) intron retentions in the context of assembly (see Section 4.4). We then develop a tool, namely irtool, based on this definition that can identify transcripts with intron retentions. This tool can be applied to an assembly generated by any assembler, allowing for the extraction or filtering out of transcripts with retained introns. Using this tool, we quantify the impact of intron retention on Scallop2 and StringTie2.

We apply irtool to filter out transcripts with (partial) intron retentions in the StringTie2 and Scallop2 assemblies, and then evaluate the filtered assemblies using RefSeq and Ensembl annotations. Based on the fact that Ensembl annotates many more transcripts with intron retention than RefSeq (Table 2), and that Scallop2 assembles more such transcripts than StringTie2 (Figs 5 and 6), we expect that the accuracy of Scallop2 would decrease under Ensembl annotation and increase under RefSeq annotation after filtering. The experimental results confirm this conjecture, as shown in Table 3. When evaluated with RefSeq, Scallop2's precision improve by 27.0% with a small decrease in recall (4.4%), while StringTie2 gains a 7.3% increase in precision but loses 4.0% in recall. When evaluated with Ensembl, Scallop2's precision increases by 14.0%, but there is a significant (14.3%) drop in recall. In contrast, StringTie2 improves by 4.8% in precision but decreases by 6.0% in recall. The changes in accuracy resulting from filtering in individual samples are shown in Figs A and B in S1 Text (GRCh38 annotations) and Figs C and D in S1 Text (T2T-CHM13 annotations). In all cases

**Table 3. Comparison of relative change in percentage of precision and the number of matching transcripts after filtering out transcripts with intron retentions in StringTie2 and Scallop2 assemblies, evaluated with different annotations as the reference.** Numbers are averaged over all samples in each dataset.

| dataset | aligner | genome | annotation | StringTie2 | | Scallop2 | |
|---------|---------|--------|------------|------------|------------|------------|------------|
| | | | | Δ prec. | Δ # mat. | Δ prec. | Δ # mat. |
| EN10 | HISAT2 | GRCh38 | RefSeq | +7.7% | -5.1% | +30.1% | -5.1% |
| EN10 | HISAT2 | T2T | RefSeq | +8.2% | -4.6% | +30.4% | -4.8% |
| EN10 | STAR | GRCh38 | RefSeq | +8.9% | -4.5% | +36.9% | -4.7% |
| EN10 | STAR | T2T | RefSeq | +9.5% | -4.1% | +37.6% | -4.5% |
| HS7 | HISAT2 | GRCh38 | RefSeq | +4.9% | -3.7% | +17.8% | -4.3% |
| HS7 | HISAT2 | T2T | RefSeq | +5.1% | -3.5% | +17.8% | -4.2% |
| HS7 | STAR | GRCh38 | RefSeq | +5.5% | -3.5% | +22.7% | -3.9% |
| HS7 | STAR | T2T | RefSeq | +5.7% | -3.3% | +23.0% | -3.8% |
| | Summary | | RefSeq | +6.9% | -4.0% | +27.0% | -4.4% |
| EN10 | HISAT2 | GRCh38 | Ensembl | +6.0% | -6.8% | +17.1% | -16.4% |
| EN10 | HISAT2 | T2T | Ensembl | +5.3% | -7.2% | +15.4% | -16.2% |
| EN10 | STAR | GRCh38 | Ensembl | +6.4% | -6.8% | +20.5% | -16.7% |
| EN10 | STAR | T2T | Ensembl | +6.4% | -6.7% | +20.8% | -16.6% |
| HS7 | HISAT2 | GRCh38 | Ensembl | +3.2% | -5.3% | +8.0% | -12.2% |
| HS7 | HISAT2 | T2T | Ensembl | +3.4% | -5.1% | +8.1% | -12.1% |
| HS7 | STAR | GRCh38 | Ensembl | +3.7% | -5.2% | +12.3% | -12.1% |
| HS7 | STAR | T2T | Ensembl | +3.9% | -5.0% | +12.4% | -11.9% |
| | Summary | | Ensembl | +4.8% | -6.0% | +14.3% | -14.3% |

(combinations of dataset, aligner, and genome build used), Scallop2 shows a sharper slope than StringTie2 on RefSeq annotations, indicating a much higher gain in precision with a similar decrease in recall.

As observed in Section 2.1, Scallop2 performs better than StringTie2 with Ensembl annotations but worse with RefSeq annotations. We now compare the accuracy of Scallop2 after filtering (referred to as Scallop2-ft) and unfiltered StringTie2. The results, presented in Table A in S1 Text show that Scallop2-ft outperforms StringTie2 with RefSeq annotations, evidenced by its outperforming on 45 (out of 68) samples. Scallop2-ft still outperforms StringTie2 with Ensembl annotations, although the margin is not as large as the one with unfiltered Scallop2 (see Table 1). We also compare the accuracy of both assemblers after filtering, i.e., String-Tie2-ft and Scallop2-ft in Table B in S1 Text. The results reveal that Scallop2-ft has a draw with StringTie2-ft with RefSeq annotations, and that Scallop2-ft outperforms StringTie2-ft on all samples.

To further evaluate the impact of partial intron retention (criterion 1 and criterion 2 of irtool, see Section 4.4) and entire intron retention (criterion 3 of irtool, see Section 4.4) on assembly accuracy separately, We compared the relative change in assembly accuracy of StringTie2 and Scallop2 on filtering partial intron retention only and filtering entire intron retention only (Table C in S1 Text). We observed that partial intron retention plays a more effective role in enhancing the precision of Scallop2's assemblies with a slightly decrease of sensitivity when evaluated with the RefSeq annotation. This observation aligns with our initial expectations. In developing the irtool, we conducted an investigation into the characteristics of transcripts annotated as "retained_intron" within the Ensembl annotation. We found that approximately one-third of "retained_intron" transcripts possessed exon(s) that overlapped with entire introns in other transcripts, a classic indicator of intron retention. Furthermore, our investigation revealed that the remaining approximately two-thirds of "retained_intron" transcripts exhibited partial intron retention. Therefore, three designed criteria help ensure the comprehensive coverage of both partial intron retention and entire intron retention.

We evaluate the performance of irtool with different threshold of minimum length-ratio on StringTie2's and Scallop2's assemblies (Fig E in S1 Text). We draw the precision-sensitivity curve to see the capability of length-ratio threshold to balance sensitivity and precision. The precision-sensitivity curves for both assemblers evaluated with annotations (Ensembl and RefSeq) show a linear relationship as length-ratio varies. This suggests that there may not be an optimal choice of length-ratio; the selection of a suitable length-ratio mainly depends on users' preference.

We then evaluate irtool with different threshold of minimum coverage-ratio on StringTie2's and Scallop2's assemblies (Fig F in S1 Text). We draw the precision-sensitivity curve to see the capability of coverage-ratio threshold to balance sensitivity and precision. To have better balance of sensitivity and precision, we usually want to choose the most top-right point (i.e. higher sensitivity and higher precision) in the precision-sensitivity curve. The points of default coverage-ratio threshold 0.5 usually located (or near) the most top-right on curves for all combinations of assemblers, annotations, aligners and datasets. This indicates the default coverage-ratio 0.5 may be a reasonable choice for general cases.

Users may choose the most suitable parameter, assembler and pipeline according to their specific requirements. For examples, if a more comprehensive assembly is preferred, particularly when transcripts with retained introns are needed, then Scallop2 may be the best choice among Scallop2-ft, StringTie2, and StringTie2-ft; on the other hand, if transcripts with retained introns are not needed, then Scallop2-ft would be the best option among the other choices.

## 2.5 More comparisons across assemblers

Some assemblers such as StringTie2 can run in annotation-guided manner by using a given transcriptome as input to boost the assembly accuracy. We perform a broader comparison across four reference-based assembly methods, including StringTie2, Scallop2, StringTie1, and StringTie2-G (i.e., annotation-guided StringTie2). The results were shown in Table D in S1 Text. StringTie2-G significantly outperforms all other three methods. This indicates that annotation-guided assembly can be beneficial when a well-annotated transcriptome is available. We note that there is a potential risk of being biased towards the provided annotation. In this experiment, the annotation provided to StringTie-G is also used in the evaluation, which may partially contribute to the improvement of StringTie2-G.

We also evaluate assemblers using the union and intersection of Ensembl and RefSeq annotations, shown in Table E in S1 Text. We observe that all assembly methods has higher precision and number of matching transcripts on the union set of annotations, compared with the corresponding numbers in Table D in S1 Text. This observation aligns with the expectation since union contains a more comprehensive set of transcripts, and suggests that evaluation with union of different annotations may be a better choice.

We now elaborate the methodological differences between StringTie and Scallop series, trying to give another perspective that explains the discrepancy when they are evaluated with RefSeq and Ensembl annotations. The algorithmic core of Scallop series, Scallop1 and Scallop2, is characterized by a "phase-preserving" strategy, ensuring all phasing paths constructed from reads, except for those identified as false positives, are comprehensively covered in the final output of assembled transcripts. This methodology is designed to to fully "respect" the nuances of reads and their alignments throughout the assembly process. In particular, when there are reads or partial of reads aligned within intron regions, Scallop2/Scallop1 consistently tries to assemble them into transcripts. Such reads overlapping with intron regions are commonly interpreted as indicative of intron retention events, resulting in transcripts with intron-retentions. On the other hand, StringTie2 series, StringTie1 and StringTie2, takes a divergent approach by employing a robust, greedy-based algorithm, which iteratively calculates the "heaviest" path in the splice graph. This approach is robust to the "noisy" reads in the intron regions, as such reads often exhibit low coverage and does not span the entire intron. As a result of this algorithmic distinction, StringTie2/StringTie1 assembles a reduced number of transcripts with intron-retentions in comparison to Scallop2/Scallop1. Ensembl annotates a much higher number of intron-retained transcripts while RefSeq rarely incorporating such transcripts. The combination of this fact and above analysis explains the divergent performance on intron-retentions when these assemblers are evaluated with different annotations.

According to Table D in S1 Text, StringTie1 exhibits reduced precision and identifies fewer matching transcripts compared to the other three methods across various combinations of aligners, datasets, annotations, and genome builds. The improvement of StringTie2 over StringTie1 may be attributed to its more aggressive strategy for identifying and removing spurious spliced alignments. Specifically, StringTie2 requires 25% more reads than StringTie1 to support spliced reads. Also, StringTie2 accepts spliced alignments with a long intron only if a at least 150% longer anchor is present on both sides of the splice site. This strict strategy of loading spliced alignments than StringTie1 likely contribute to its improvement.

## 3 Discussion

In this work we assess the impact of annotations on transcript assembly. We have discovered that different conclusions can arise when using different annotations for evaluation. To unravel this mystery, we analyzed the transcript biotypes across various annotations and

assemblies, and figured that the bias in annotating and assembling transcripts with retained introns is the main cause.

Moreover, we investigated similarities and differences in annotations at the intron-exon boundary, junction, and intron-chain levels, and our results indicate that the primary structural divergences in annotations occur at the intron-chain level. In addition, we have developed a standalone tool for extracting or filtering out transcripts with retained introns from an assembly, freely available at https://github.com/Shao-Group/irtool. This tool can be used in conjunction with any assembler to mitigate bias in transcript assembly with intron retentions. Our results show that the accuracy improvement varies significantly when applying this tool to different assemblers. Specifically, we found that Scallop2-ft (Scallop2 followed by filtering) is superior to Scallop2, StringTie2, and StringTie2-ft when producing an assembly without intron-retained transcripts.

## 4 Methods

### 4.1 Evaluation pipeline

We evaluate Scallop2 (version 1.1.2), StringTie1 (version 1.3.6), StringTie2 (version 2.2.0), with and without `-G` option which enables the annotation-guided assembly. All assemblers were run with default parameter setting. The command lines of running StringTie2 and StringTie1 are the same: `stringtie <input.bam> -o <output.gtf>`. When an annotation is provided, StringTie2 use the following command line to perform annotation-guided assembly: `stringtie <input.bam> -G <reference.gtf> -o <output.gtf>`. We use `scallop2 -i <input.bam> -o <output.gtf>` to run Scallop2. The assembled transcripts (from any method) are evaluated with GffCompare by command line: `gffcompare -r <reference.gtf> -o <outout.prefix> <query.gtf>`. The irtool is run by `irtool <input.gtf> <intron-retention. gtf> <filtered.gtf>`.

We use two human RNA-seq datasets: EN10, consisting of 10 paired-end RNA-seq samples downloaded from the ENCODE project, and HS7, containing 7 paired-end RNA-seq samples used in the Long Read Genome Annotation Assessment Project. The assembled transcripts are assessed mainly using five transcriptome annotations derived from two human genome builds: GRCh38 and T2T-CHM13. GRCh38 is the most commonly used human genome assembly, while T2T-CHM13 is the most recent and comprehensive, gapless sequence of a human genome. For GRCh38, we use the latest Ensembl annotation (release 107 on genome build GRCh38.p13) and RefSeq annotation (release 110 on genome build GRCh38.p14). We use 3 annotations from T2T-CHM13, namely the Ensembl annotation, RefSeq annotation, and its own CHM13 annotation released with its paper.

We created two additional annotations, the union and intersection sets of Ensembl and RefSeq annotations, by GffCompare. These two sets were also employed to evaluate the assembly accuracy of different assembly methods. Specifically, we run GffCompare by taking the Ensembl annotation as reference and RefSeq annotation as query. GffCompare generates a `. tmap` output file for the query RefSeq annotation. The `.tmap` file includes a tag `Class Code` for each transcript in the query annotation. We say a transcript $t$ is in the set of intersection set of Ensembl and RefSeq annotations if its `Class Code` is "=". We collect all transcripts in the RefSeq annotation that have a `Class Code` as "=" to form the intersection set. We collect all transcripts in the RefSeq annotation that does not have a `Class Code` as "=" to form a unique set, then union it with the Ensembl annotation to form the final union set.

We use the pipeline depicted in Fig 7 to assess the assemblers' accuracy using different annotations. Each RNA-seq sample is aligned with two popular splice-aware aligners, STAR

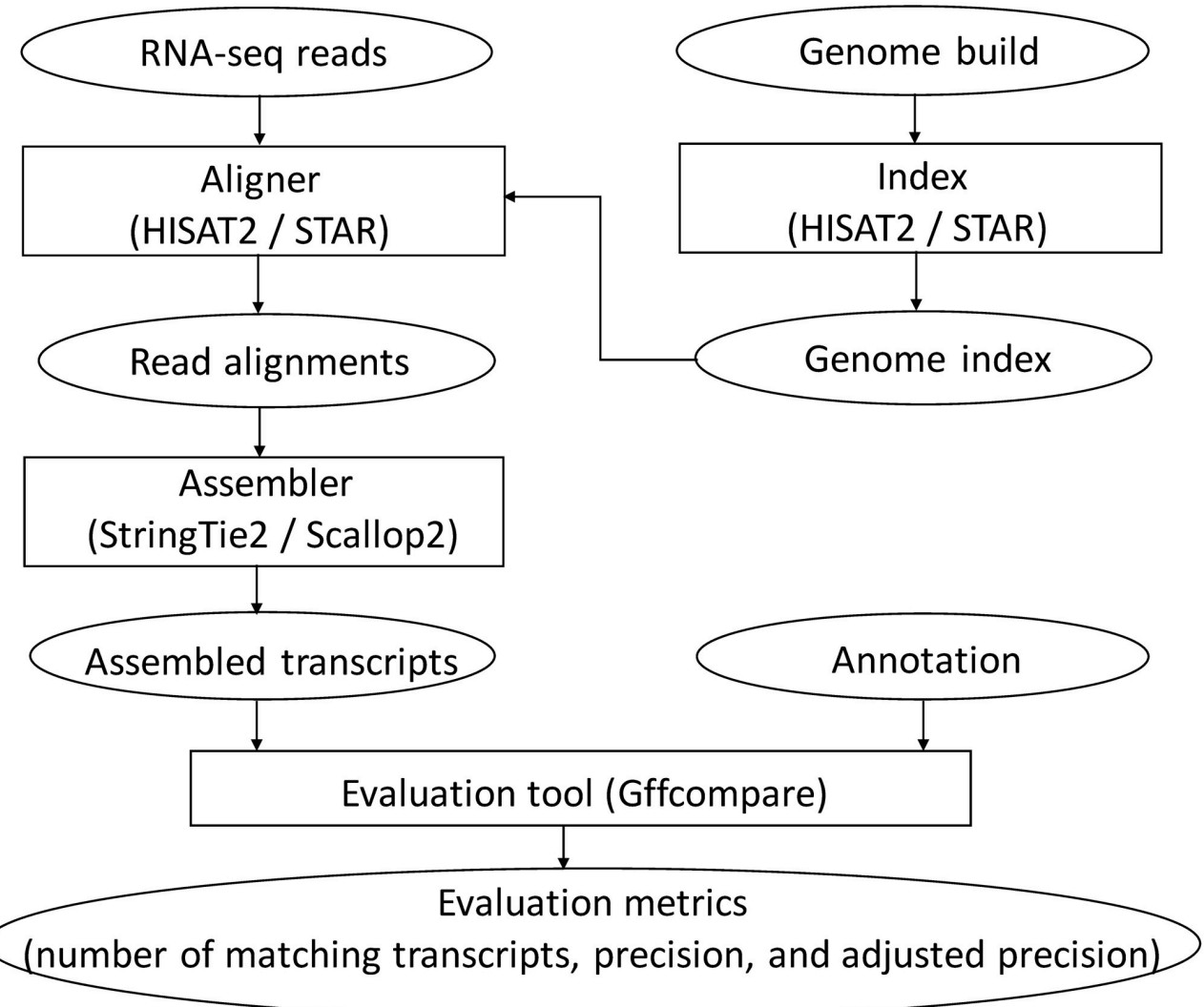

**Fig 7. Pipeline of evaluating the accuracy of compared assemblers.**

[33] and HISAT2 [34]. The resulting read alignments will be piped to the assemblers, producing a set of assembled transcripts. The accuracy of the assembled transcripts will be evaluated using tool gffcompare [35], with one of the annotations serving as the ground-truth. We use the "transcript level" measure defined by gffcompare: an assembled multi-exon transcript is considered to be "matching" if its intron-chain exactly matches that of a transcript in the annotation; an assembled single-exon transcript is defined as "matching" if there is a significant overlap (80% by default) with a single-exon transcript in the annotation. We report the two metrics calculated by gffcompare: the total number of matching transcripts, which is proportional to recall, and precision, defined as the total number of matching transcripts divided by the total number of assembled transcripts. On samples where mixed outcomes exhibit, i.e., one method performs better on one metric but not on the other, we compare their *adjusted precisions*, defined as their precisions when matching transcripts are adjusted to be the same. Specifically, we take the assembly with higher recall and gradually remove its transcripts with lowest (predicted) abundance. In this process its recall will drop but its precision will likely

increase as abundance is highly correlated with accuracy. We step this process when its recall matches that of the other assembler, and the precision at this moment (i.e., the adjusted precision) will be compared with the precision of the other assembler. This way of comparison has been used in previous studies [28, 30].

## 4.2 Metrics for structural similarities

We propose a set of metrics for measuring the structural similarities of two annotations. Let $t$ be a transcript. We use $B(t)$, $J(t)$, and $C(t)$ to represent the set of intron-exon boundaries, the set of junctions, and the intron-chain, of $t$, respectively. Let $g$ be a gene, which may contain multiple transcripts in an annotation, we define $B(g) = \cup_{t \in g} B(t)$, $J(g) = \cup_{t \in g} J(t)$, and $C(g) = \cup_{t \in g} C(t)$, to represent the set of boundaries, junctions, and intron-chains, of all transcripts in gene $g$, respectively. Let $T$ be an annotation with many annotated genes. We then define $B(T) = \cup_{g \in T} B(g)$, $J(T) = \cup_{g \in T} J(g)$, and $C(T) = \cup_{g \in T} C(g)$. We use *Jaccard similarity* to measure the structural similarity of two annotations. Formally, let $T_1$ and $T_2$ be two annotations, we use $J_B(T_1, T_2) := |B(T_1) \cap B(T_2)|/|B(T_1) \cup B(T_2)|$, $J_J(T_1, T_2) := |J(T_1) \cap J(T_2)|/|J(T_1) \cup J(T_2)|$, and $J_C(T_1, T_2) := |C(T_1) \cap C(T_2)|/|C(T_1) \cup C(T_2)|$, to measure the similarity of $T_1$ and $T_2$ at the level of boundary, junction, and intron-chain, respectively. Please see Fig 8 for an illustration of these definitions.

## 4.3 Constructing gene correspondence

We focus on multi-exon genes when constructing the correspondence between genes in two annotations, i.e., genes annotated with at least one multi-exon transcript. We propose a simple approach: two genes $g_1 \in T_1$ and $g_2 \in T_2$, where $T_1$ and $T_2$ are two annotations, form a pair if they share at least one intron-exon boundary, i.e., $B(g_1) \cap B(g_2) \neq \emptyset$. Note that in this definition one gene in $T_1$ may form gene pairs with multiple genes in $T_2$, but this rarely happens since two genes in one annotation normally do not share intron-exon boundaries.

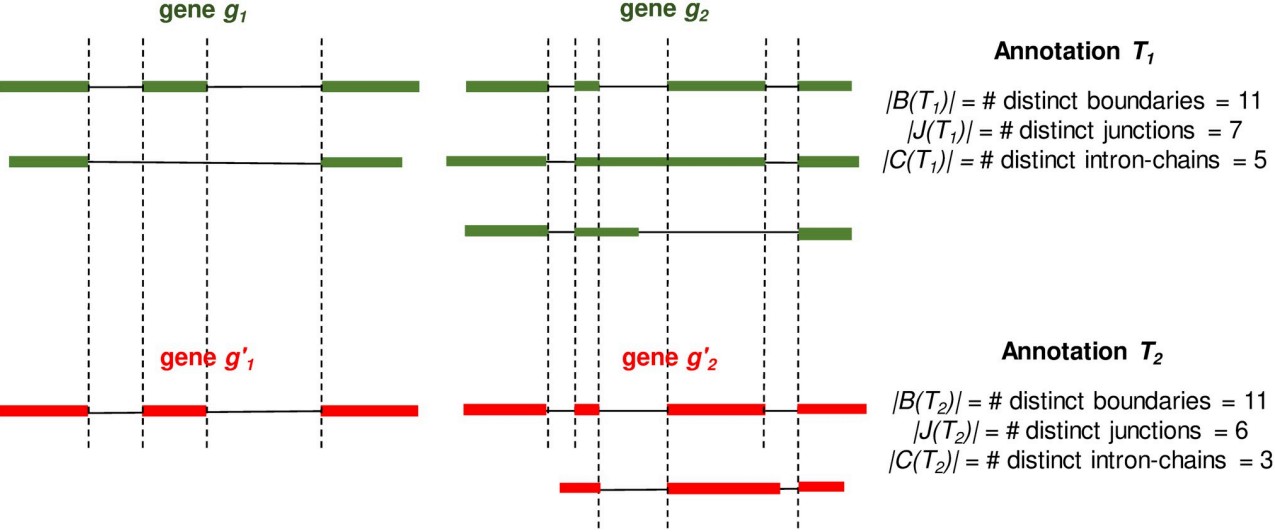

**Fig 8. A toy example for illustrating the Jaccard similarity of two annotations $T_1$ and $T_2$ at the level of boundary, junction, and intron-chain.** Genes and transcripts from the same annotation are colored the same. Identical boundaries between two annotations are marked with vertical dashed lines. We have $J_B(T_1, T_2) := |B(T_1) \cap B(T_2)|/|B(T_1) \cup B(T_2)| = 5/6|$, $J_J(T_1, T_2) := |J(T_1) \cap J(T_2)|/|J(T_1) \cup J(T_2)| = 5/8|$, and $J_C(T_1, T_2) := |C(T_1) \cap C(T_2)|/|C(T_1) \cup C(T_2)| = 1/3|$.

The Jaccard similarity of each constructed gene pair $(g_1, g_2)$ at boundary, junction, and intron-chain levels, can be defined similarly, formally written as $J_B(g_1, g_2) := |B(g_1) \cap B(g_2)|/|B(g_1) \cup B(g_2)|$, $J_J(g_1, g_2) := |J(g_1) \cap J(g_2)|/|J(g_1) \cup J(g_2)|$, and $J_C(g_1, g_2) := |C(g_1) \cap C(g_2)|/|C(g_1) \cup C(g_2)|$.

## 4.4 Definition of intron retentions in the context of assembly

We describe our definition of (partial) intron retentions. To determine if a transcript $t$ in an assembly has intron retention or not, we need to find another transcript $r$ in the same assembly as reference, and compare $t$ with $r$. The definition also uses the abundances (i.e., expression levels) of $t$ and $r$; we therefore assume that each transcript in the assembly are associated with an abundance. Most assemblers, including StringTie2 and Scallop2, assembles transcripts while also predicting their abundances. Let $p(t)$ and $p(r)$ be the abundances of transcripts $t$ and $r$, respectively. We define transcript $t$ has *intron retention* if there exists transcript $r$ (in the same assembly with $t$) such that $p(r)/p(t)$ is above a threshold (a parameter termed *coverage ratio*; 0.5 by default) and either (a) the first exon of $t$ spans an intron and the following exon of $r$ (Fig 9, criterion 1), or (b) the last exon of $t$ spans an exon and the following intron of $r$ (Fig 9, criterion 2), or (c) there exists an exon in $t$ and an intron in $r$ such that the intron is fully covered by the exon (Fig 9, criterion 3). Note that one transcript $t$ may satisfy two or more criteria with the same $r$, or may satisfy one or more criteria with different $r$.

We refer to the criteria 1 and 2 defined above as *partial intron retention* and criterion 3 as *entire intron retention*. irtool provides two options, option `-po <bool>` to turn off partial intron retention and keep entire intron retention only, i.e. turn off criterion 1 and 2, keep criterion 3 only, and option `-wo <bool>` to turn off entire intron retention and keep partial intron retention, i.e. turn off criterion 3 but keep criterion 1 and 2.

In order to characterize partial intron retention more precisely, irtool provides two parameters, *coverage ratio* as defined above, and *length ratio*, defined as the the ratio between the length of the first/last exon that overlaps with the intron and the length of the intron and 0 by

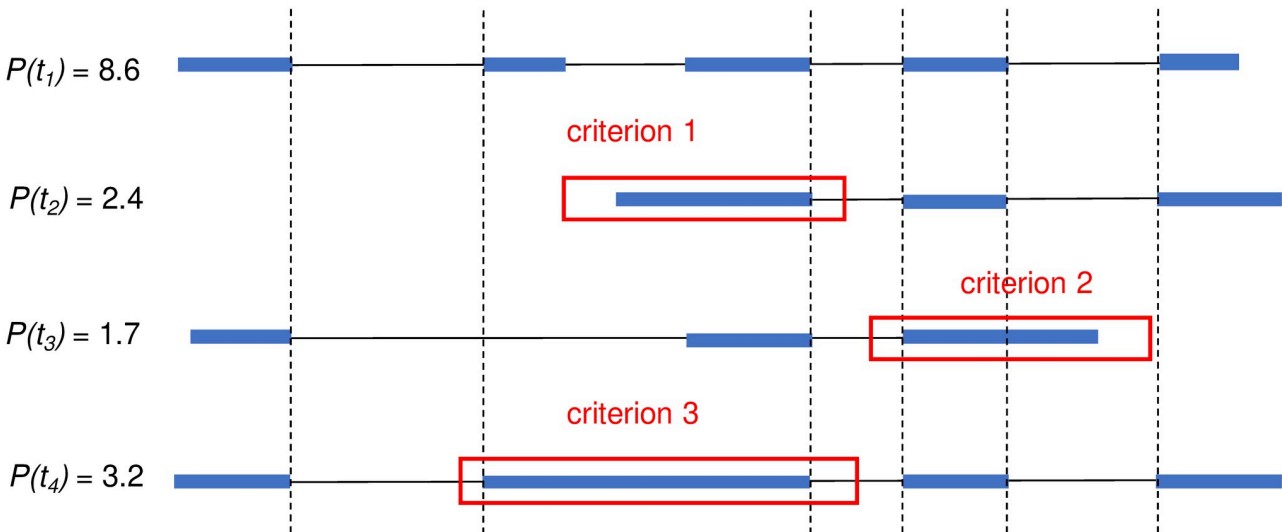

**Fig 9. An illustrative example for the three criteria used to define transcripts with intron retentions.** Identical boundaries are marked with vertical dashed lines. Transcript $t_2$ satisfies criterion 1 (it has lower abundance than $t_1$, and its first exon spans the second intron of $t_1$). Transcript $t_3$ satisfies criterion 2 (it has lower abundance than $t_1$ and its last exon spans the fourth intron of $t_1$). Transcript $t_4$ satisfies criterion 3 (it has lower abundance than $t_1$ and its second exon fully covers the second intron of $t_1$).

default. Adjusting either one provides a trade-off between sensitivity and precision. More number of transcripts can be identified as transcripts with intron retention when the length-ratio or coverage-ratio becomes lower. Hence, the sensitivity decreases while the precision increases when the length-ratio or coverage-ratio gets lower. Users can specify their preferred choice of coverage ratio with option `-cr <double>` and length ratio with option `-lr <double>`.

## Supporting information

**S1 Text. Supplementary materials including Figs A-F, Tables A-E.**
(PDF)

**S1 Data. Containing all numerical data used in Figs 1–6.**
(XLSX)

## Acknowledgments

We would like to thank Troy LaPolice for his work on exploratory experiments.

## Author Contributions

**Conceptualization:** Qimin Zhang, Mingfu Shao.

**Formal analysis:** Qimin Zhang, Mingfu Shao.

**Funding acquisition:** Mingfu Shao.

**Investigation:** Qimin Zhang, Mingfu Shao.

**Methodology:** Qimin Zhang, Mingfu Shao.

**Project administration:** Mingfu Shao.

**Software:** Qimin Zhang, Mingfu Shao.

**Supervision:** Mingfu Shao.

**Visualization:** Qimin Zhang, Mingfu Shao.

**Writing – original draft:** Qimin Zhang, Mingfu Shao.

**Writing – review & editing:** Qimin Zhang, Mingfu Shao.

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
