## [Decision Letter · Decision Letter 0]

25 Sep 2023

Dear Dr. Shao,

Thank you very much for submitting your manuscript "Transcript Assembly and Annotations: Bias and Adjustment" for consideration at PLOS Computational Biology.

As with all papers reviewed by the journal, your manuscript was reviewed by members of the editorial board and by several independent reviewers. In light of the reviews (below this email), we would like to invite the resubmission of a significantly-revised version that takes into account the reviewers' comments.

I am sorry for the time this took, there were a total of 22 reviewers invited to obtain these two reviews.

We cannot make any decision about publication until we have seen the revised manuscript and your response to the reviewers' comments. Your revised manuscript is also likely to be sent to reviewers for further evaluation.

Sincerely,

Marc Robinson-Rechavi

Academic Editor

PLOS Computational Biology

Jason Papin

Editor-in-Chief

PLOS Computational Biology

Reviewer's Responses to Questions

**Comments to the Authors:**

Reviewer #1: In the manuscript “Transcript Assembly and Annotations: Bias and Adjustment”, the authors investigated the influence of different transcript annotations (RefSeq and Ensembl) on evaluating the accuracy of transcriptome assembly tools. Based on the observation that using different annotations to benchmark the assembly tools can lead to confliction outcomes, the authors first compared the structural similarity between different annotations, and then investigated the transcript biotypes in different annotations and revealed that the differences between annotations are mainly related to intron retentions. Then, the authors proposed a tool to adjust the biases in intron retention for the assembled transcripts and provided a guidance on how to choose the suitable assembler for the specific requirements.

To some extent, the research done by the authors make sense, since no criteria exists for deciding which annotation should be used to evaluate the transcriptome assemblers for now. And, different groups generally used different annotations to benchmark the assembler, e.g., in StringTie paper, the combination of the three annotations RefSeq, UCSC, and Ensembl was used as the ground truth, StringTie2 paper used only the RefSeq annotation, Scallop and Scallop2(developed by the authors of this manuscript) paper used the Ensembl annotation, PsiClass[PMID: 31676772] paper used the GENCODE annotation, and TransMeta[PMID: 35858749] paper used both RefSeq and GENCODE annotations, etc.

Overall, it appears to be a manuscript with rational methodology design and carefully-performed evaluation, while I still have some concerns as follows.

1. During the development of an algorithm, a specific transcriptome annotation library was generally used to train the computational model, and training with different annotations would greatly affect the tendency of the final assembly. This maybe an important cause to result in confliction outcomes when evaluating the assemblers with different annotations. I suggest the authors analyze the phenomenon from the root cause, rather than just from the final assembled results.

2. As stated by the authors, Scallop2 identified much more intron retained transcripts than StringTie2. It seems that StringTie2 removes some kind of the aligned reads from the alignment files generated by Hisat2 or Star. Please the authors make more analysis from the methodology of StringTie2 and Scallop2, and this maybe provide new ideas for the development of new assemblers. Moreover, will StringTie1 have a similar performance as StringTie2? Since a great difference between StringTie2 and StringTie1 is StringTie2’s more aggressive strategy for identifying and removing spurious spliced alignments.

3. Will it be good strategies to use the union or intersection of different transcriptome annotations as the ground truth to benchmark the transcriptome assembly tools?

4. According to the author’s investigation, the annotation-guided assembly way (taking as input an annotation library to guide the assembly procedure) looks like a good strategy. And, StringTie2 has such an option, while Scallop2 doesn’t. Please the authors discuss this.

5. The figure 3 and 4 are referred to before figure 1 and 2 in the manuscript, figure numbers should fit citation’s order, please the authors rearrange.

6. In page 24, section 4.3, “... form a pair if they share at least one intron-exon boundary, i.e., J(g1) ∩ J(g2) …” should be “…i.e., B(g1) ∩ B(g2)..”.

Reviewer #2: In this manuscript, the authors explored and identified potential biases when evaluating transcriptome assembly methods. The authors compared Scallop2 and StringTie2, two popular reference-guided transcriptome assemblers, on ten RNA-seq data, using two sets of genome annotations from RefSeq and Ensembl on hg38 and CHM13, respectively. The authors observed that Scallop2 and StringTie2’s accuracy comparisons were inconsistent when using RefSeq and Ensemble annotations. The authors identified that one of the main reasons could be due to the intron retention transcripts. After removing qualified intron retention transcripts, the authors found Scallop2 obtained consistently higher precision and sensitivity in both annotation systems. The evaluations in this manuscript are well-designed, and the findings are important for future transcriptome assembler method development and benchmarks. However, there are several concerns the authors shall address before the manuscript can be published on PLOS Computational Biology.

1. Partial intron retention should be split into two categories. Based on Figure 13, criterion 1 and criterion 2 may correspond to alternative transcription start/end sites. Traditionally, only criterion 3 should be considered intron retention as an alternative splicing event. The authors shall compare Scallop2 and StringTie2 on criterion1/2 and criterion 3 separately, which may provide more insights into the behaviors of the two transcriptome assemblers and the differences between RefSeq and Ensembl annotations.

2. For Figure 13’s criterion1/2, except for the abundance ratio threshold, is there a length cutoff? For example, only when the exon of r covers more than 50% of the intron from another transcript t, r can be regarded as an intron retention transcript.

3. The default value for determining intron retention transcript is the p(r)/p(t)=0.5. Fold-2 change is a reasonable choice, but since one of the major findings in this manuscript is the differences in intron retention transcripts, the authors may need more justifications for this threshold. One suggestion is to draw precision-#matching-transcripts curves for StringTie2-ft and Scallop2-ft when varying the p(r)/p(t) threshold.

4. Even though the intron retention transcripts caused the evaluation discrepancy, ignoring them at all might be too drastic, as they could be real. The authors may consider using long reads or other evidence to confirm that intron retention transcripts are common in a sample. In other words, RefSeq annotation might be too conservative, and Ensembl annotation is a better representation of the transcripts, especially for intron retention transcripts. This might help establish a better guideline for future transcriptome assembler evaluations.

5. One minor concern is that the manuscript may have too many figures and tables. Though these figures are very informative, they may become distractions. For example, Figure 7-10 are large figure panels, and they are for underlying sample-level changes for Table 3. The authors may consider putting some of Figure 7-10 in supplementary figures to make the manuscript more concise.

Another example is Table 4, which is the comparison between the original StringTie2 output and the post-processed Scallop2-ft output. The information can be directly extracted from Table 1 and Table 5. Furthermore, comparing post-processed Scallop2-ft with an unprocessed StringTie2’s would be deemed as an unfair comparison. Meanwhile, Table 5’s comparison between StringTie2-ft and Scallop2-ft is a fairer comparison.

**Have the authors made all data and (if applicable) computational code underlying the findings in their manuscript fully available?**

Reviewer #1: Yes

Reviewer #2: **No: **Though the authors have released the code for the irtool, the evaluation codes, such as running command of gffcompare, StringTie2 and Scallop2, in the manuscript do not seem publicly available.

PLOS authors have the option to publish the peer review history of their article (what does this mean?). If published, this will include your full peer review and any attached files.

Reviewer #1: **Yes: **Guojun Li

Reviewer #2: No
---

## [Editor Report · Decision Letter 1]

20 Nov 2023

Dear Dr. Shao,

Thank you very much for submitting your manuscript "Transcript Assembly and Annotations: Bias and Adjustment" for consideration at PLOS Computational Biology.

I have checked your revision, and I see that all changes of consequence were made in supplementary materials. As most readers will read only the main manuscript, this means that they will not have access to these revisions, which are an important result of the work of the reviewers. Thus I am asking you to please incorporate these revisions within the main text, after which I will send the revised manuscript to the reviewers for their further consideration.

We cannot make any decision about publication until we have seen the revised manuscript and your response to the reviewers' comments. Your revised manuscript is also likely to be sent to reviewers for further evaluation.

Sincerely,

Marc Robinson-Rechavi

Academic Editor

PLOS Computational Biology

Jason Papin

Editor-in-Chief

PLOS Computational Biology
---

## [Decision Letter · Decision Letter 2]

4 Dec 2023

Dear Dr. Shao,

We are pleased to inform you that your manuscript 'Transcript Assembly and Annotations: Bias and Adjustment' has been provisionally accepted for publication in PLOS Computational Biology.

(Please note you can address the minor comment with submission of final manuscript files.)

Best regards,

Marc Robinson-Rechavi

Academic Editor

PLOS Computational Biology

Jason Papin

Editor-in-Chief

PLOS Computational Biology

Reviewer's Responses to Questions

**Comments to the Authors:**

Reviewer #1: Authors answered all the questions I raised.

Reviewer #2: The authors have addressed all my concerns. There is one minor unclear point in the revised manuscript:

The manuscript uses both “full intron retention” and “entire intron retention” terms. Are their definitions interchangeable? It would better to use one of them for consistency purpose.

**Have the authors made all data and (if applicable) computational code underlying the findings in their manuscript fully available?**

Reviewer #1: Yes

Reviewer #2: Yes

PLOS authors have the option to publish the peer review history of their article (what does this mean?). If published, this will include your full peer review and any attached files.

Reviewer #1: **Yes: **Guojun Li

Reviewer #2: No

---

## [Editor Report · Acceptance letter]

15 Dec 2023

PCOMPBIOL-D-23-01081R2 

Transcript Assembly and Annotations: Bias and Adjustment

Dear Dr Shao,

I am pleased to inform you that your manuscript has been formally accepted for publication in PLOS Computational Biology. Your manuscript is now with our production department and you will be notified of the publication date in due course.

With kind regards,

Judit Kozma
